# Prevalence and Risk Factors for Renal Insufficiency among Adults Living with HIV in Tanzania: Results from a Cross-Sectional Study in 2020–2021

**DOI:** 10.3390/healthcare12060657

**Published:** 2024-03-14

**Authors:** Theresia A. Ottaru, Gideon P. Kwesigabo, Zeeshan Butt, Matthew Caputo, Pilly Chillo, Hellen Siril, Lisa R. Hirschhorn, Claudia Hawkins

**Affiliations:** 1Department of Epidemiology and Biostatistics, Muhimbili University of Health and Allied Sciences, Dar es Salaam P.O. Box 65001, Tanzania; gkwesigabo@muhas.ac.tz; 2Phreesia, Inc., Wilmington, DE 19803, USA; zbutt@phreesia.com; 3Department of Psychiatry and Behavioral Sciences, Feinberg School of Medicine, Northwestern University, Chicago, IL 60611, USA; 4Robert J. Havey Institute for Global Health, Feinberg School of Medicine, Northwestern University, Chicago, IL 60611, USA; matthew.caputo@northwestern.edu (M.C.); lisa.hirschhorn@northwestern.edu (L.R.H.); c-hawkins@northwestern.edu (C.H.); 5Department of Internal Medicine, Muhimbili University of Health and Allied Sciences, Dar es Salaam P.O. Box 65001, Tanzania; pchillo@muhas.ac.tz; 6Department of Psychiatry and Mental Health, Muhimbili University of Health and Allied Sciences, Dar es Salaam P.O. Box 65001, Tanzania; hsiril@mdh.or.tz; 7Department of Medical Social Sciences, Feinberg School of Medicine, Northwestern University, Chicago, IL 60611, USA

**Keywords:** renal insufficiency, kidney disease, age, HIV infection, comorbidities, Tanzania

## Abstract

With improved survival, adults living with HIV (ALHIV) are increasingly likely to experience age-related and HIV-related comorbidities, including renal insufficiency. Other risk factors for renal insufficiency (high blood pressure (BP), obesity, diabetes, and dyslipidemia) are also growing more common among ALHIV. To determine the prevalence of renal insufficiency (defined as an eGFR < 60 mL/min/1.73 m^2^) and factors associated with reduced eGFR, we conducted a cross-sectional study at six HIV clinics in Dar-es-Salaam, Tanzania. We applied multivariable (MV) ordinal logistic regression models to identify factors associated with reduced eGFR and examined the interaction of age with BP levels. Among the 450 ALHIV on ART analyzed [26% males; median age 43 (IQR: 18–72) years; 89% on tenofovir-containing ART; 88% HIV viral load ≤50 copies/mL], 34 (7.5%) had renal insufficiency. Prevalence was higher among males (12%) vs. females (6%), *p* = 0.03; ALHIV ≥50 (21%) vs. <50 years (2.5%), *p* < 0.001; those with high [≥130/80 mmHg (15%)] vs. normal [<120/80 mmHg (4%)] BP, *p* < 0.01 and those with dyslipidemia (10%) vs. those without (4.5%), *p* < 0.03. After adjusting for covariates, age (in years) was the only covariate with a statistically significant association with reduced eGFR (OR = 1.09 (1.07–1.12), *p* < 0.001). No significant interaction between age and BP was found. Interventions to increase routine screening for renal insufficiency, especially among older ALHIV, and improve BP control are critical to reducing kidney disease-related morbidity and mortality.

## 1. Introduction

With the development of effective combination antiretroviral therapy (ART), the mortality rate from HIV infection has decreased significantly among adults living with HIV (ALHIV) [1,2,3]. As survival has improved, the aging population of ALHIV is now at increased risk for developing age-related and HIV-related comorbidities, including kidney disease [4]. Kidney disease presenting as renal insufficiency is regarded as one of the major factors contributing to high morbidity and mortality among ALHIV [5,6].

In comparison to persons without HIV, ALHIV have a nearly four-fold increased likelihood of developing renal insufficiency, even after adjusting for traditional risk factors such as high blood pressure (BP) and diabetes mellitus [7]. Globally the prevalence of renal insufficiency, defined as an estimated glomerular filtration rate (eGFR) of <60 mL/min/1.73 m^2^, among ALHIV ranges between 3 and 7% [8]. In Sub-Saharan Africa (SSA), where over two-thirds of the world’s population of ALHIV reside [3], the pooled prevalence of renal insufficiency among ALHIV is reported to be as high as 12% [8,9]. Renal insufficiency is also common among ALHIV in Tanzania, with prevalence ranging from 7 to 11% [10,11,12,13].

Studies have documented several mechanisms through which HIV infection increases the risk of renal insufficiency. These include long-term use of nephrotoxic ART (tenofovir, indinavir, and lopinavir/ritonavir) [14,15] and direct renal cell injury from HIV infection associated chronic inflammation [16]. Other HIV infection-related risk factors include low CD4+ T cell counts, high HIV viral loads (VLs), and coinfection with hepatitis C and/or Hepatitis B virus [17]. ALHIV are also at risk for renal insufficiency from traditional risk factors [11,12,17], including older age, high BP, diabetes mellitus, dyslipidemia, obesity, and smoking, which are increasingly common among ALHIV in Tanzania [13,18,19,20,21,22,23,24]. Studies from the US have shown that persons of African descent are also at increased risk for renal insufficiency and disease due to polymorphisms in the apolipoprotein L1 (APOL1) gene [25,26].

Older age is a key predictor of renal insufficiency among both adults with and without HIV. Several studies have associated the aging process with molecular and cellular damage, which results in reduced eGFR. The synergistic role of age-related, HIV-related, and traditional risk factors with renal insufficiency remains largely unexplored in ALHIV. In this study, we determined the prevalence of renal insufficiency and risk factors associated with reduced eGFR among ALHIV in Tanzania. Furthermore, we examined age and its interaction with BP levels. High BP is a crucial risk factor for renal insufficiency that significantly contributes to its progression to end-stage renal disease (ESRD) [27] and is highly prevalent among ALHIV [28].

## 2. Materials and Methods

### 2.1. Study Design and Setting

This cross-sectional study included ALHIV attending six high volume (>1500 clients) HIV care and treatment clinics (CTCs) in five districts of Dar es Salaam, Tanzania, who were enrolled between November 2020 and January 2021 in a study examining cardiovascular health and other outcomes [18].

### 2.2. Study Participants

Eligible participants were adults ages ≥18 years on ART who had been attending one of the six HIV CTCs for ≥12 months prior to the start of the study. Participant selection is described in [18]. Participants who were pregnant or unable to give informed consent were excluded.

### 2.3. Current Standard of Care at the CTC

Since 2019, Tenofovir disoproxil fumarate + Lamivudine + Dolutegravir (TDF + 3TC + DTG) has been the recommended first-line regimen for ALHIV in Tanzania. Other regimens are recommended according to HIV National guidelines (MOH, 2019). TDF has been a component of the first-line ART regimens in Tanzania since 2012. By 2017, the year Tanzania adopted a “treat all” approach, the majority of ALHIV were on TDF. Prior to ART initiation, all clients have serum creatinine and urinalysis for proteinuria performed. Clients with creatinine clearance <60 mL/min or dipstick proteinuria of ≥1 undergo further investigation (including renal ultrasound) to determine the extent of impaired renal function and are started on renal specialized care outside the HIV clinic and a non-TDF based regimen such as Abacavir (ABC) + 3TC + DTG [29].

During follow-up, clients on TDF-based regimens are recommended by the guideline to have serum creatinine and dipstick proteinuria performed every six months and screening for other kidney disease risk factors, including BP (every visit), blood glucose (every six months), and lipids (yearly). The extent to which these recommendations are implemented varies across the HIV CTCs, and it largely depends on the availability of equipment and support from the implementing partners. 

### 2.4. Data Collection

Data collected included sociodemographic and clinical factors including age and gender, health insurance status, smoking status, alcohol use, type and duration on ART, BP, height, weight (to determine BMI), and laboratory data including serum creatinine, random blood glucose (RBG), and lipid profile. Blood tests were performed during data collection. Further details of the study and laboratory procedures have been published elsewhere [18]. Participants who were found to have raised values of BP, RBG, or total cholesterol were referred to the outpatient clinic of the internal medicine department at their respective health facility for management.

### 2.5. Variables

The primary outcome was renal insufficiency, defined as an eGFR of <60 mL/min/1.73 m^2^ according to the KDIGO 2012 Clinical Practice Guideline for the Evaluation and Management of Chronic Kidney Disease (CKD) [30]. eGFR was estimated from serum creatinine using the 2021 CKD-EPI equation and was used as a continuous outcome in the regression analysis. The 2021 CKD-EPI formula is a race-free formulae that has been widely used to assess renal insufficiency among Africans in SSA [11,12]. Use of race-based CKD-EPI has been shown to overestimate eGFR if applied among African populations, consequently resulting in delayed initiation of nephrological care including dialysis initiation [31,32,33]. The eGFR values were also categorized according to the KDIGO 2012 Clinical Practice Guideline [30]. The categories (in mL/min/1.73 m^2^) include normal/high (≥90), mildly decreased (60–89), mild–moderately decreased (45–59), moderately–severely decreased (30–44), severely decreased (15–29), and kidney failure (<15). Participants were considered smokers and alcohol users if they were currently smoking or drinking, or they quit in ≤12 months [34].

Blood pressure was categorized as normal (<120/80 mmHg), elevated (120–129/<80 mmHg), and high (≥130/80 mmHg and/or use of antihypertensive therapy) according to the American Heart Association (AHA) hypertension guidelines [35]. Studies have demonstrated that the risk of kidney damage starts to increase when the BP is ≥130/80 mmHg, especially among individuals with other comorbidities such as obesity [36]. Participants were asked to sit comfortably for at least five minutes before the measurement. BP measurement included three values taken at least five minutes apart, which were averaged to determine the final value [35]. Diabetes mellitus was defined as an RBG value of ≥11.1 mmol/L and/or use of antidiabetic medication [37], while dyslipidemia was defined as having either high total cholesterol (≥5.2 mmol/L), high triglycerides (≥1.7 mmol/L), high low-density lipoprotein cholesterol (LDL-C) (≥3.4), or low high-density lipoprotein cholesterol (HDL-C) (<1.0 among males and <1.3 among females), as per the National Cholesterol Education Program Adult Treatment Panel III (ATP III) [38] and as used in previous studies in this population [39,40].

### 2.6. Data Analysis

Data analysis was conducted using STATA version 14 (STATA Corp Inc., College Station, TX, USA). We calculated descriptive statistics using the median (interquartile range (IQR)) for continuous variables and frequencies (percentages) for categorical variables stratified by the presence or absence of renal insufficiency, with *p*-values calculated by univariable logistic regression. We also computed the proportion of participants in each eGFR stage based on the KDIGO criteria, categorized by age (<50 vs. ≥50 years), and reported the *p*-value for the chi-square test for trend. 

We applied multivariable (MV) ordinal logistic regression models to identify factors associated with reduced eGFR. The three eGFR outcome levels used in this analysis were normal/high (≥90), mildly decreased (60–89), and moderately to severely decreased (renal insufficiency) (<60). In the MV model, we included all covariates that we believed could have been associated with eGFR or were important to adjust for based on the previous literature, regardless of their *p*-values in univariable analysis. The exception to this was the 6-leveled participant facility variable, which was not intended to be included in the multivariable analysis unless univariable associations were observed with the outcome, in order to minimize overfitting with our limited sample size. Diabetes mellitus was not included in the model due to its low prevalence in our sample. The variables age, BMI, and years lived with HIV infection were treated continuously. Multicollinearity was assessed with generalized variance inflation factors (GVIFs), with (GVIF^1/(2DF)^)^2^ > 3, where DF represents degrees of freedom, indicating consideration for removal from the model. We tested the proportional hazards assumption for all covariates with a Brant test, graphically assessing any variables with *p* < 0.05.

Considering that both age and BP levels are significant risk factors for renal insufficiency, and with the high prevalence of high BP among ALHIV and the increasing proportion of aging ALHIV, we were interested in assessing the combined role of age and BP levels. Thus, an interaction term between age and BP was introduced into the MV model but was only to be included in the final model if *p* < 0.05. This interaction was also represented graphically, using age as a binary variable (<50 years and ≥50 years), and 95% confidence intervals (CIs) for stratified prevalence of renal insufficiency were computed with the normal approximation. All analyses were two-tailed with the significance level set at 5%.

## 3. Ethical Consideration

The study was approved by Muhimbili University of Health and Allied Sciences (MUHAS)-MUHAS-REC-08-2020-343, the National Institute for Medical Research (NIMR)-NIMR/HQ/R8a/VOL.IX/3513, and Northwestern University (STU00214283) ethics committees. All participants provided written informed consent to participate. 

## 4. Results

### 4.1. Characteristics of the Study Population

Of the 629 participants enrolled in the original cohort [18], 179 (28.5%) had missing values for serum creatinine (consequently missing eGFR values) due to invalid laboratory results, mislabeling, or declining blood draw, and were therefore excluded from the analysis. The remaining 450 ALHIV [median age of 43 (IQR 18–72) years, 74% females] were included in this analysis (see Table 1).

The median (IQR) eGFR was 96 (81–109) mL/min/1.73 m^2^; 7.5% (n = 34) had eGFR < 60 mL/min/1.73 m^2^. Further stratifying by eGFR categories (see Figure 1), 63% (n = 284) had eGFR > 90 mL/min/1.73 m^2^; 29% (n = 132) had 60–89 mL/min/1.73 m^2^; 6% (n = 27) had 45–59 mL/min/1.73 m^2^; 1.3% (n = 6) had 30–44 mL/min/1.73 m^2^; 0.2% (n = 1) had 15–29 mL/min/1.73 m^2^. None of the participants in our study population had kidney failure (eGFR < 15 mL/min/1.73 m^2^).

Most participants were on a TDF-based regimen (89.1%), with a median ART duration of 5 years (IQR 3–10 years) and had HIV VL levels ≤ 50 copies/mL (88%) (see Table 1). A quarter of the study population had high BP (26%) and were currently using alcohol (25%), while 24% were obese. The prevalence of renal insufficiency was higher among males, ALHIV ≥ 50, those with elevated and high vs. normal BP, and those with dyslipidemia (all with *p* < 0.05) (see Table 1).

### 4.2. Distribution of eGFR Stage According to KDIGO 2012 Criteria by Age

Fewer ALHIV ≥ 50 years had normal/high renal function according to the KDIGO criteria. The prevalence of all stages of reduced eGFR (from mildly to severely decreased renal function) was significantly higher among ALHIV ≥ 50 years (chi-square test for trend, *p* < 0.001) (see Figure 1).

### 4.3. Associations with Reduced eGFR

In unadjusted ordinal logistic regression models, age (OR = 1.09 (1.07–1.12), *p* < 0.001), high BP level (OR = 2.18 (1.38–3.44), *p* < 0.001), and dyslipidemia (OR = 2.18 (1.39–3.42), *p* < 0.001) were significantly associated with reduced eGFR (see Table 2). 

All covariates of interest satisfied the ordinal logistic regression assumptions and were included in the MV model, except for the facility variable, which was deemed unlikely to be a confounder and dropped to avoid overfitting. Six participants were excluded from the model due to missing values. After adjusting for sex, BP levels, dyslipidemia, years lived with HIV infection, ART regimen, and HIV VL copies, age (in years) was the only covariate with a statistically significant association with reduced eGFR (OR = 1.09 (1.07–1.12), *p* < 0.001). A positive association between alcohol use and reduced eGFR was found, but this was not statistically significant (OR = 1.56 (0.95–2.54), *p* = 0.06). 

The interaction term for age and BP was not statistically significant (*p* = 0.80) and was therefore removed from the final model. The prevalence of renal insufficiency by age and BP levels (combined) is shown in Figure 2. The prevalence of renal insufficiency was about seven times higher among ALHIV aged ≥50 years compared to ALHIV < 50 years across the BP levels.

## 5. Discussion

In this study, we assessed the prevalence, risk factors, and potential synergistic role of age and BP on reduced eGFR among ALHIV. We found a relatively low prevalence of renal insufficiency in a population of ALHIV on ART for a median of 5 years, mostly TDF-containing ART. Although the prevalence of renal insufficiency using the eGFR < 60 mL/min/1.73 m^2^ definition was low, more than a third of ALHIV had below normal renal function (eGFR < 90 mL/min/1.73 m^2^). Only age had a statistically significant association with reduced eGFR. 

The prevalence of renal insufficiency in this study is comparable to other studies among ALHIV within Tanzania, which reported a prevalence of renal insufficiency (eGFR < 60 mL/min/1.73 m^2^) of 7–10% among ALHIV in Tanzania [10,11,12,13]. In general, studies from SSA [8,9,10,11,12] have reported much higher rates of renal insufficiency (7–12%) among ALHIV, using similar definitions (eGFR < 60 mL/min/1.73 m^2^), compared to studies from central-eastern Europe [41] and the Asian-pacific region [42,43,44] (4–5%), although the settings and populations within the studies were highly variable. Systematic reviews in the US have reported a prevalence of renal insufficiency of up to 15.5% among ALHIV [45,46]. More than half of the participant population in such studies have been African Americans living with HIV. These findings suggest an underlying genetic role in the development and progression of renal function loss, as previously documented [47].

The relatively high proportion of participants with reduced renal function (eGFR < 90 mL/min/1.73 m^2^) without insufficiency is also of concern. This population may be at risk for further eGFR decrease while on TDF. As this study was only cross-sectional, we have no way of determining the role of long-term use of TDF in these declines. TDF is an antiretroviral associated with an accelerated decrease in eGFR, proximal tubular dysfunction, and chronic kidney disease [48]. Studies have shown that the risk of renal toxicity from TDF use is higher among those with pre-existing renal dysfunction or at a higher risk of renal dysfunction due to comorbidities such as high BP and diabetes [47,49]. Tanzania faces an epidemiological transition that is associated with an increase in the burden of noncommunicable diseases (NCDs) including hypertension [50], which places ALHIV at additional risk as they age. In fact, over one-half of our population was found to have elevated BP. A lack of resources, including diagnostic equipment, in Tanzania has significantly hindered the implementation of recommended routine screening and monitoring of renal functioning among ALHIV [51]. Longitudinal studies in Tanzania on (1) the effect of long-term TDF exposure in this population, particularly as they age and (2) interventions to increase the rates of routine screening and monitoring for renal insufficiency in HIV CTCs, are of paramount importance. 

In this study, age was a strong predictor of reduced eGFR, a finding similar to other studies in SSA [10,11,12]. Studies have shown that chronic HIV infection accelerates aging and, even among ALHIV well controlled on ART, there is still an increased risk of age-related renal function loss [26]. Currently, about 15% of ALHIV in SSA are aged ≥50 years. Prediction models estimate that the proportion will increase to 27%, equivalent to about 9.1 million older ALHIV, by 2040 [52]. While the prevalence of renal insufficiency reported in this study and other studies within SSA is relatively low, the increase in the cumulative number of individuals aging with HIV infection will undoubtedly result in a significant rise in the number of people at risk of, and living with, kidney disease in SSA. Thus, strategies for prevention, early detection, and management of renal insufficiency are critical in this population.

A quarter of the study population had high BP levels (BP of ≥130/80 mmHg or on antihypertensive therapy). In both age groups (<50 years and ≥50 years) the prevalence of renal insufficiency was two times greater for those with high BP compared to those with normal or elevated BP. In response to the growing burden of NCDs, including hypertension, the Tanzania Ministry of Health prepared the National NCDs Strategic Plan (2016–2020) [53]. Initiatives introduced to control the burden of hypertension include (1) awareness campaigns on regular BP screening, risk factors associated with high BP, and management; (2) health system strengthening through training of providers on the management of high BP; and (3) integration of high BP services within primary health care, including HIV CTCs, to ensure accessibility of screening and treatment services for high BP [53]. Although there are plans to monitor and evaluate the progress of the strategic plan, current data on the progress of implementation, including the provision of integrated hypertension services within HIV CTCs in Tanzania, are not available. Findings from our study suggest that efforts to prevent and manage high BP among ALHIV in the HIV CTCs are vital to prevent renal insufficiency and will largely depend on the success of these initiatives. In this study, we did not see a significant interaction between age and BP levels on renal insufficiency, contrary to our expectations. Larger studies are needed to confirm this finding in Tanzania and similar settings as the aging HIV population continues to expand.

There are some limitations to our study. First, the definition of renal insufficiency was limited to a single value of reduced eGFR of <60 mL/min/1.73 m^2^ and lacked the contribution of proteinuria. As a result, the prevalence of renal insufficiency reported in this study could have underestimated the prevalence of renal insufficiency among ALHIV in Tanzania. However, research has shown that even a single eGFR < 60 mL/min/1.73 m^2^ is an important predictor of increased mortality and cardiovascular risk [54,55], thus, the findings from this study could be a meaningful first step in understanding the burden of renal insufficiency in the at-risk population of ALHIV in Tanzania. Secondly, missing values for creatinine levels limited our analysis to a restricted sample of a subset of participants from our cohort (n = 450, 72%). This could result in an underestimation of the true prevalence of renal insufficiency in our study population if patients with higher rates were excluded from the analysis. However, there were no differences in important characteristics associated with renal insufficiency (such as median age, mean SBP, and DBP) between the analytical sample and those excluded. Due to our limited sample size, we did not include the facility variable, a potential confounder, as a random effect in our multivariable analysis. Participants were not asked to fast prior to data collection, as a result, the proportion of participants with diabetes mellitus and dyslipidemia reported in this study could be overestimated. Finally, because of the cross-sectional design of this study, the assessment of chronic renal insufficiency from declines in renal function meant that cases could have been either acute or chronic.

## 6. Conclusions

In conclusion, the prevalence of renal insufficiency using the KDIGO 2012 criteria was low among ALHIV. However, more than a third of participants showed evidence of mild declines in renal function, putting them at risk of further declines as they age and develop age-related comorbidities. Our results highlight the need for regular assessments of renal function among ALHIV, particularly older ALHIV. Additionally, research is needed to identify and test interventions to improve screening for and management of renal insufficiency among older ALHIV to improve overall health outcomes.

## Figures and Tables

**Figure 1 healthcare-12-00657-f001:**
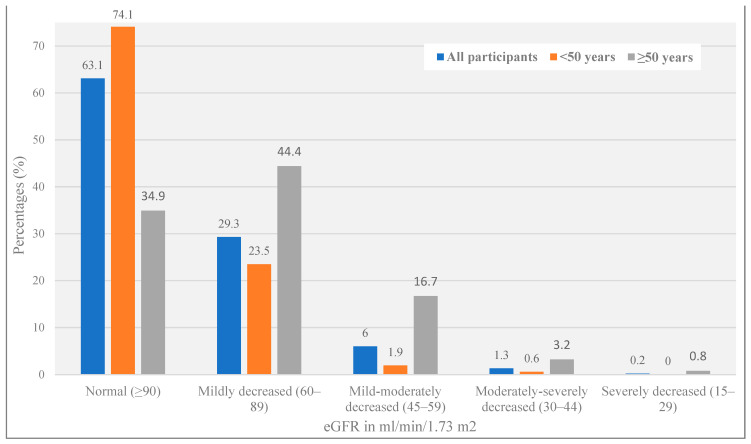
Distribution of the study population based on eGFR stage, according to the KDIGO criteria, by age.

**Figure 2 healthcare-12-00657-f002:**
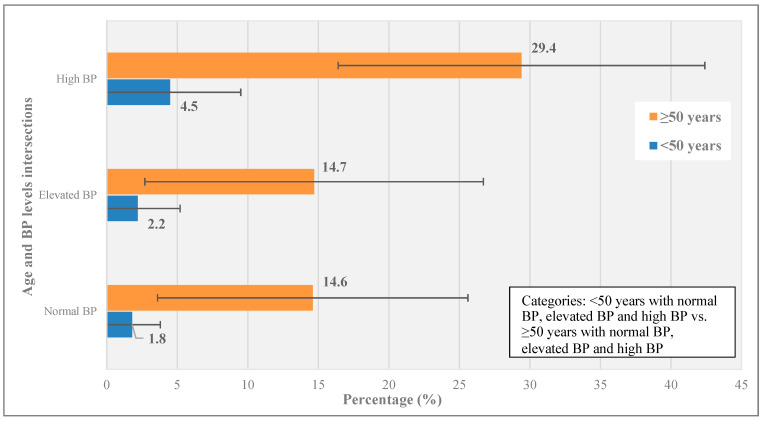
Prevalence of renal insufficiency by age (<50 vs. ≥50 years) and BP level (normal, elevated, and high) categories.

**Table 1 healthcare-12-00657-t001:** Characteristics of the study population.

Characteristic		Renal Insufficiency (eGFR < 60 mL/min/1.73 m^2^)		
	Total (N = 450)	Yes (n = 34)	No (n = 416)	*p*-Value
Demographics				
	n (%) ^a^	n (%) ^b^	n (%) ^b^	
Age (in years)				
Below 50	324 (72.0)	8 (2.5)	316 (97.5)	Ref.
50 and above	126 (28.0)	26 (20.6)	100 (79.4)	<0.001
Sex				
Female	332 (73.8)	20 (6)	312 (94)	Ref.
Male	118 (26.2)	14 (11.9)	104 (88.1)	0.03
Health insurance status				
No	383 (85.1)	26 (6.8)	357 (93.2)	Ref.
Yes	67 (14.9)	8 (11.9)	59 (88.1)	0.15
Clinical characteristics				
BP levels (mm/Hg) *				
Normal (BP < 120/80)	206 (45.8)	9 (4.4)	197 (95.6)	Ref.
Elevated (BP 120–129/<80)	127 (28.2)	7 (5.5)	120 (94.5)	0.64
High (BP ≥ 130/80) or on antihypertensive therapy	117 (26.0)	18 (15.4)	99 (84.6)	0.001
Diabetes (RBG ≥ 11.1 mmol/L) or on antidiabetic therapy *				
No	437 (97.1)	32 (7.32)	405 (92.7)	Ref.
Yes	13 (2.9)	2 (15.4)	11 (84.6)	0.28
Total Cholesterol (mmol/L)				
<5.2	376 (84)	20 (5.3)	356 (94.7)	Ref.
≥5.2	74 (16)	14 (18.9)	60 (81.1)	<0.001
Triglycerides (mmol/L)				
<1.7	373 (82.8)	21 (5.6)	352 (94.4)	Ref.
≥1.7	77 (17)	13 (16.9)	64 (83.1)	0.001
LDL-C (mmol/L)				
<3.4	380 (84)	21 (5.5)	359 (94.5)	Ref.
≥3.4	70 (16)	13 (18.6)	57 (81.4)	<0.001
HDL-C (mmol/L)				
≥1.0 among male and ≥1.3 among Female	268 (59.6)	19 (7)	249 (93)	Ref.
<1.0 among Male and <1.3 among Female	182 (40.4)	15 (8.2)	167 (91.8)	0.65
Dyslipidemia				
No	201 (45)	9 (4.5)	192 (95.5)	Ref.
Yes	249 (55)	25 (10)	224 (90)	0.03
BMI (kg/m^2^)				
Normal weight (<25)	228 (50.7)	17 (7.5)	211 (92.5)	Ref.
Overweight (25–30)	114 (25.3)	8 (7)	106 (93)	0.88
Obese (≥30)	108 (24.0)	9 (8.3)	99 (91.7)	0.78
Smoking status				
No	377 (83.8)	27 (7.2)	350 (92.8)	Ref.
Yes	73 (16.2)	7 (9.6)	66 (90.4)	0.47
Alcohol use				
No	338 (75.1)	24 (7.1)	314 (92.9)	Ref.
Yes	112 (24.9)	10 (8.9)	102 (91.1)	0.53
HIV infection related characteristics				
Years lived with HIV infection (median, IQR) in years	6 (3–11)	6.5 (3–11)	6 (3–11)	0.50
Duration on ART (median, IQR) in years	5 (3–10)	6 (3–10)	5 (3–10)	0.49
ART regimen				
TDF + 3TC + DTG (TLD)	401 (89.1)	32 (8)	369 (92)	Ref.
Other regimens	49 (10.9)	2 (4)	47 (96)	0.33
HIV VL load				
≤50 copies	394 (87.6)	32 (8.1)	362 (91.9)	Ref.
>50 copies	56 (12.4)	2 (3.6)	54 (96.4)	0.23

^a^ Column percentages and ^b^ row percentages. ART—antiretroviral therapy; BMI—body mass index; BP—blood pressure; HDL-C—high-density lipoprotein cholesterol; IQR—interquartile range; LDL-C—low-density lipoprotein cholesterol; TDF—Tenofovir; 3TC—Lamivudine; DTG—Dolutegravir; TLD—Tenofovir, Lamivudine, and Dolutegravir; RBG—random blood glucose; VL—viral load. Smoking status: Yes—current or quit smoking within the last 12 months; No—not a current smoker or quit smoking >12 months ago. Alcohol use: Yes—current or quit drinking ≤12 months; No—not currently drinking or quit drinking >12 months. Dyslipidemia—high total cholesterol (≥5.2 mmol/L), or high triglycerides (≥1.7 mmol/L), or high LDL-C (≥3.4), or low HDL-C (<1.0 among males and <1.3 among females). *p*-value—univariable logistic regression. * Missing values—2.2% for BP levels and 0.4% for RBG levels. Mean imputation method was used to impute for missing values.

**Table 2 healthcare-12-00657-t002:** Risk factors associated with reduced eGFR.

Characteristic	Crude OR (95% CI) (N = 444)	*p*-Value	Adjusted OR (95% CI) (N = 444)	*p*-Value
Age (in years)	1.09 (1.07–1.12)	<0.001	1.09 (1.07–1.12)	<0.001
Sex				
Female	Ref.		Ref.	
Male	1.51 (0.99–2.30)	0.05	1.25 (0.74–2.10)	0.35
BP levels (mmHg)				
Normal (BP < 120/80)	Ref.		Ref.	
Elevated (BP 120–129/<80)	1.04 (0.65–1.65)	0.87	0.93 (0.55–1.55)	0.25
High (BP ≥ 130/80) or on antihypertensive therapy	2.18 (1.38–3.44)	<0.001	1.28 (0.77–2.13)	0.31
Dyslipidemia				
No	Ref.		Ref.	
Yes	2.18 (1.39–3.42)	<0.001	1.55 (0.94–2.54)	0.15
BMI (kg/m^2^)	0.99 (0.96–1.00)	0.29	0.98 (0.94–1.00)	0.24
Smoking status				
No	Ref.			
Yes	0.92 (0.54–1.53)	0.74	0.62 (0.32–1.17)	0.15
Alcohol use				
No	Ref.			
Yes	1.16 (0.75–1.78)	0.50	1.56 (0.95–2.54)	0.06
Years lived with HIV infection (in years)	1.03 (0.99–1.07)	0.21	0.98 (0.94–1.03)	0.33
ART regimen				
TDF + 3TC + DTG (TLD)	Ref.		Ref.	
Other regimens	0.64 (0.33–1.20)	0.18	0.78 (0.36–1.59)	0.53
HIV VL (copies/mL)				
≤50 copies	Ref.		Ref.	
>50 copies	0.63 (0.34–1.14)	0.14	0.63 (0.31–1.22)	0.19
Facility				
Site 1	1.31 (0.72–2.41)	0.38		
Site 2	Ref.			
Site 3	1.58 (0.88–2.85)	0.13		
Site 4	1.19 (0.65–2.20)	0.57		
Site 5	1.03 (0.52–2.02)	0.92		
Site 6	0.92 (0.44–1.86)	0.82		

ART—antiretroviral therapy; BMI—body mass index; BP—blood pressure; HDL-C—high-density lipoprotein cholesterol; IQR—interquartile range; LDL-C—low-density lipoprotein cholesterol; TDF—Tenofovir; 3TC—Lamivudine; DTG—Dolutegravir; TLD—Tenofovir, Lamivudine, and Dolutegravir; RBG—random blood glucose; VL—viral load. Smoking status: Yes—current or quit smoking ≤12 months; No—not a current smoker or quit smoking >12 months. Alcohol use: Yes—current or quit drinking ≤12 months; No—not currently drinking or quit drinking >12 months. Dyslipidemia—high total cholesterol (≥5.2 mmol/L), or high triglycerides (≥1.7 mmol/L), or high LDL-C (≥3.4), or low HDL-C (<1.0 among males and <1.3 among females). *p*-value—UV and MV ordinal logistic regression.

## Data Availability

Data are contained within the article.

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
