# Peer review of "Prevalence and Risk Factors for Renal Insufficiency among Adults Living with HIV in Tanzania: Results from a Cross-Sectional Study in 2020–2021"

_healthcare, 2024, doi:10.3390/healthcare12060657_

Round 1

Reviewer 1 Report (Previous Reviewer 1)

Comments and Suggestions for Authors

The authors invested considerable effort to shift from a logistic to a linear regression model as the primary way of analysis. Nevertheless, I still have three concerns with the study in its revised form:

1.       Screening interactions should either be restricted to pairs of variables that were selected based on clinical/biological pre-knowledge that certain effects might be modulated by other effects (in the present case only by age), or on a full screening (each predictor x age, with appropriate multiple testing). Testing only interactions of a certain factor with those variables that were (as main effects) significant in the multi-variable model without interactions might lead to wrong conclusions, since interactions might be significant even though their main effects (in a model without interactions) are not.

2.       It is standard to include center in models for multi-center study data. A random effect is a parsimonious way to do so which would result in a linear mixed model (random intercept model) in this case. If the authors have investigated a potential center effect but it proved neither statistically significant nor having relevant influence on the estimates, they should report this.

3.       As already discussed, the conclusion “…suggesting that management of hypertension is necessary regardless of age” might be based on a false negative result of the interaction test. Even though the authors recommend larger studies for exactly this research question I think this conclusion is premature. Maybe a more cautious formulation might suffice: At the current state of research (including the present study) we cannot exclude that… or we need to assume that…

Author Response

Reviewer 2 Report (Previous Reviewer 3)

Comments and Suggestions for Authors

I thank the authors for providing a revised version of their manuscript, which has been slightly improved.

However, I still have some concerns, specifically in such a retrospective study :

-       I think that it’s not possible to be sure that all the blood pressure measurements, have been done according to the recommandations,

-       What kind of medical recors are used to collect the informations, and to be sure they are accurate ?

-       About sugar : although the cutt-off of 11,1 mmol/l is used, I think that fasting results are better ; in addition, did all the patients have a blood test at the time they have been included ? which would be rather surprising … were older values used ?

-       Same comments about lipid values.

Finally, even with some improvements, I don’t really see any new value of the results presented.

Comments on the Quality of English Language

no specific comment

Author Response

Reviewer 3 Report (New Reviewer)

Comments and Suggestions for Authors

Ottaru et al. investigated the prevalence of renal insufficiency defined as eGFR<60ml/min/1.73m2) and factors associated with reduced eGFR in people with HIV in Tanzania. They used linear regression.

The manuscript is well written and should be published.

I however have a major methodological issue.

eGFR is not a really linear variable. For example, value of 100 is not worse than value of 130, but value of 58 is much worse than value of 90.  This means that , when authors conduct linear regression and show the change of , for example, 10 units , these can be change from eGFR  60 to 50 but also from eGFR 130 to 120.  That’s being said, linear regression is not an appropriate method in such studies.

Authors presented in Table 1 variables for patients with eGFR <60 vs >=60; I would conduct logistic regression with eGFR <60 as dependent variable instead of linear regression. In this case, analyses will be methodologically appropriate.

Round 2

Reviewer 3 Report (New Reviewer)

Comments and Suggestions for Authors

Thank you for adressing my comments

This manuscript is a resubmission of an earlier submission. The following is a list of the peer review reports and author responses from that submission.

Round 1

Reviewer 1 Report

Comments and Suggestions for Authors

The authors present an interesting study of prevalence and risk factor finding that is well reported and clearly structured. I have some concerns regarding the statistical analysis and some additional minor points to make in the following.

1.       In the abstract the authors suggest that they examined the interaction of each of the considered risk factors with age. However, in the text they only seem to look at the interaction of BP with age. This should be consistent and transparent. If more than one interactions were screened the manner of screening should be made explicit.

2.       There is some template text at the end of the section “Current standard of care” that should be removed.

3.       While missingness of the outcome (renal insufficiency) is well documented and discussed there is no information about missing values of the considered potential risk factors. If there are missing values: how did the authors deal with them? There number should be reported in Table 1.

4.       p.3, l. 130: There seems to be a category between “current” and “quit <= 12 months”, something like “quit during last year”?

5.       The abbreviation IQR should be explained at first appearance.

6.       My major concern is about the multivariable logistic regression model. Although wide-spread and common among epidemiological and clinical studies, it is well known that a pre-selection of risk factors using results of the univariable models is inadequate (see, e.g., DOI: 10.1002/bimj.201700067). Further, the events-per-variable (see same reference) is fairly low in this dataset which raises concerns of overfit. Even if the aim of this study apparently is not to build some kind of risk score, the low number of “events” (renal insufficiency) should be taken into account when building the multivariable model. Maybe, the (apparently) central question of age and BP (was this focus chosen a priori or is it data-driven?) should be answered using a very parsimonious model.

7.       Does the center cluster variable enter the model as fixed or random effect? A fixed effect would maybe be more appropriate if only a special selection of centers within the country is considered. However, this would induce five degrees of freedom that are not “available” due to the low number of events. Was the cluster effect also used in the “unadjusted” analysis?

8.       p.4, l.160: the “of” should be deleted.

9.       Fig.1: Give a hint in the caption that here the KDIGO criteria are used.

10.   First paragraph page 6: While for the categorical factors the Chi-square test tests the null hypothesis of equal prevalences (since this test is symmetrical with regard to dependent and independent variable) this is not true for the Mann-Whitney U-test (which has equal distributions of the continuous or ordinal variable between two groups as null hypothesis). Univariable logistic or Poisson models would be more appropriate here.

11.   p.7, l.199: “no interaction” is probably not true, the authors probably mean “no statistically significant interaction”. Since difference of significance is not significance of difference, the authors should report the p-value of the interaction test instead of the two p-values of the sliced effect. Further, the non-significant interaction has a high chance to be a false negative result, since statistical power is probably quite low for the interaction test. Nevertheless, it would help to give an interpretation, since many clinicians in my experience do not know what a sign. or non-sign. interaction really means.

Reviewer 2 Report

Comments and Suggestions for Authors

- Although dipstick proteinuria was checked, it was mentioned in the results, How many patients had different grades of proteinuria?

- With ART known to cause crystal mediated disease in the kidney, one of the other limitation is not checking for hematuria also.

Reviewer 3 Report

Comments and Suggestions for Authors

The authors present a retrospective study, conducted in 6 HIV clinics in Dar es Salaam, Tanzania, evaluating the prevalence and risk factors for renal insufficiency, in a cohort of adult HIV-infected patients, on ARV treatment. Overall prevalence of eGFR < 60 ml/mn is of 7,5%, with only age > 50 years and high blood pressure being significantly associated in a multivariate model. Although interesting, the study is retrospective, and some information are difficult to be correctly, and accurately, collected (diabetes, other medical conditions, fasting or not for blood results); in addition, the presented results doesn’t really add any new value.

METHODOLOGY

-       I don’t understand the period of recruitment (even if ref 18 indicates some details, it should indicated in the present paper).

-       Blood pressure: even if AHA categories have been used, I’m not convinced by these scale, and was wondering how blood pressure has been measured (AHA, for example, recommends to control the values before concluding); there is also the question of white coat high blood pressure.

-       Ref 36 doesn’t define diabetes mellitus; in addition, were the blood sugar measurements done while fasting, or not ? was it possible to find this information in a retrospective study ? therefore, was diabetes correctly diagnosed (and not over or underestimated) ?

-       Ref 37 doesn’t define hypercholesterolemia; same as above, were the patients fasting ? are there any LDL data ? Diagnosis of hypercholesterolemia appears to be difficult, and is also dependent on other CV risk factors.

-       Specific HIV factors are not taken into account: duration of HIV positivity, initial HIV viral load, duration of viral control (below detection limit), duration of HIV-treatment, which HIV drugs in the past or present combination, and of course duration of TDF … It is difficult, in my mind, to study renal insufficiency in HIV-infected patients, without adding TDF in the model … 

-       Were other comorbidities taken into account (cardiac ischemic disease, cancers, nephrotoxic drugs, …) ?

-       Line 100: there is a text coming from the guidelines for authors …

REFERENCES

Ref 7 is incomplete

Comments on the Quality of English Language

English is of good quality.